

# Identification and characterization of critical genes associated with tamoxifen resistance in breast cancer

Kai Zhang[1],[*], Kuikui Jiang[2],[*], Ruoxi Hong[2], Fei Xu[2], Wen Xia[2], Ge Qin[2], Kaping Lee[2], Qiufan Zheng[2], Qianyi Lu[2], Qinglian Zhai[2] and Shusen Wang[2]

[1] Key laboratory of Carcinogenesis and Translational Research (Ministry of Education/Beijing), Laboratory of Molecular Oncology, Peking University Cancer Hospital & Institute, Beijing, China

[2] Sun Yat-sen University Cancer Center, State Key Laboratory of Oncology in South China, Collaborative Innovation Center for Cancer Medicine, Guangzhou, China

[*] These authors contributed equally to this work.

Corresponding author
Shusen Wang,
wangshs@sysucc.org.cn

## ABSTRACT

**Background:** Tamoxifen resistance in breast cancer is an unsolved problem in clinical practice. The aim of this study was to determine the potential mechanisms of tamoxifen resistance through bioinformatics analysis.

**Methods:** Gene expression profiles of tamoxifen-resistant MCF-7/TR and MCF-7 cells were acquired from the Gene Expression Omnibus dataset GSE26459, and differentially expressed genes (DEGs) were detected with R software. We conducted Gene Ontology (GO) and Kyoto Encyclopedia of Genes and Genomes pathway enrichment analyses using Database for Annotation, Visualization and Integrated Discovery. A protein–protein interaction (PPI) network was generated, and we analyzed hub genes in the network with the Search Tool for the Retrieval of Interacting Genes database. Finally, we used siRNAs to silence the target genes and conducted the MTS assay.

**Results:** We identified 865 DEGs, 399 of which were upregulated. GO analysis indicated that most genes are related to telomere organization, extracellular exosomes, and binding-related items for protein heterodimerization. PPI network construction revealed that the top 10 hub genes—*ACLY, HSPD1, PFAS, GART, TXN, HSPH1, HSPE1, IRAS, TRAP1*, and *ATIC*—might be associated with tamoxifen resistance. Consistently, RT-qPCR analysis indicated that the expression of these 10 genes was increased in MCF-7/TR cells comparing with MCF-7 cells. Four hub genes (*TXN, HSPD1, HSPH1* and *ATIC*) were related to overall survival in patients who accepted tamoxifen. In addition, knockdown of HSPH1 by siRNA may lead to reduced growth of MCF-7/TR cell with a trend close to significance ($P = 0.07$), indicating that upregulation of HSPH1 may play a role in tamoxifen resistance.

**Conclusion:** This study revealed a number of critical hub genes that might serve as therapeutic targets in breast cancer resistant to tamoxifen and provided potential directions for uncovering the mechanisms of tamoxifen resistance.

## INTRODUCTION

About 70% of breast cancer cases worldwide are estrogen receptor (ER) positive. ER-positive breast cancer is an endocrine-dependent disease, and ER plays a vital role in the metastasis, occurrence and progression of this disease (*Baum et al., 2002*; *Torre et al., 2015*; *Turner et al., 2017*). Tamoxifen, a selective estrogen receptor modulator, binds to ERα and significantly suppresses the estrogen-induced growth of mammary epithelial cells (*Early Breast Cancer Trialists' Collaborative Group, 1998*). However, approximately one-third of ER-positive breast cancers inevitably develop tamoxifen resistance, which leads to an unfavorable prognosis and remains a crucial clinical challenge to effective treatment (*Ring & Dowsett, 2004*). Thus, it is critical to elucidate the underlying factors that cause tamoxifen resistance, identify improved treatment strategies, and develop effective regimens for this disease (*Shang & Brown, 2002*; *Gutierrez et al., 2005*).

In recent years, many curative measures attempting to overcome tamoxifen resistance have been explored (*Ring & Dowsett, 2004*; *Viedma-Rodriguez et al., 2014*), and previous investigations have highlighted the use of specific molecular targeted drugs (*Normanno et al., 2005*). However, due to the unclear mechanisms, analyses of such approaches are still in the early stages and the treatment effects are not satisfactory (*Nass & Kalinski, 2015*; *Zhao et al., 2017*). Overall, the molecular mechanisms of tamoxifen resistance have not been determined. Exploration into the underlying factors of tamoxifen resistance would affect targeting strategies that might overcome tamoxifen resistance and enhance clinical results (*Rondon-Lagos et al., 2016*). Accordingly, identification of the genes related to tamoxifen resistance with powerful gene sequencing technologies, which may elucidate the mechanisms of tamoxifen resistance and help identify new curative strategies, is urgently needed (*Katzenellenbogen & Frasor, 2004*). However, few studies have been done on the key genes associated with tamoxifen resistance in breast cancer through bioinformatics analysis.

Gene microarray technology is an enhanced high-throughput method to effectively analyze gene expression profiles. Thus, gene microarrays have been comprehensively used to explore the underlying regulatory networks involved in different types of cancer and have important clinical applications in improving clinical diagnoses and discovering new drug targets (*Zhang et al., 2017*). This technology can also identify thousands of differentially expressed genes (DEGs) related to different biological processes (BPs), cellular components (CCs), molecular functions (MFs), and different signaling pathways of cancer. Moreover, bioinformatics methods allow for comprehensive analysis of large amounts of data derived from microarrays. Given the false positives and heterogeneity of different microarray results, we processed one microarray dataset, GSE26459, to obtain DEGs between the ER-positive breast cancer cell line MCF-7 (cell line sensitive to tamoxifen) and MCF-7/TR (cell line with acquired tamoxifen resistance). These findings combined with the bioinformatics results identified important signaling pathways and hub genes involved in breast cancer tamoxifen resistance. Through survival analysis, we found that *ATIC, HSPD1, HSPH1* and *TXN* were related to survival. Further knockdown of target genes by siRNAs indicated that upregulation of HSPH1 is likely to

play a role in tamoxifen resistance. Taken together, our results provide a valuable perspective into the mechanisms of tamoxifen resistance, identify possible candidate biomarkers, and suggest potential therapeutic drugs for overcoming tamoxifen resistance.

## MATERIALS AND METHODS

### Cell culture

MCF-7 and tamoxifen-resistant MCF-7 (MCF-7/TR) cell lines were obtained from the Breast Tumor Center, Sun Yat-sen Memorial Hospital, Sun Yat-sen University (*Zhu et al., 2018*). Tamoxifen-resistant MCF-7 (MCF-7/TR) cells were established as formerly reported (*Badia et al., 2000*; *Knowlden et al., 2003*). MCF-7 cells were cultured in DMEM (Gibco, Gaithersburg, MD, USA) with 10% FBS (Gibco, Gaithersburg, MD, USA). Tamoxifen-resistant MCF-7 (MCF-7/TR) cells were cultured in DMEM (Gibco, Gaithersburg, MD, USA) supplemented with 10% FBS (Gibco, Gaithersburg, MD, USA) and one μM tamoxifen.

### Identification of DEGs

To identify DEGs in tamoxifen-resistant MCF-7/TR cells and tamoxifen-sensitive MCF-7 cells, we obtained raw public gene expression data (GSE26459) from the Gene Expression Omnibus (GEO) database. This dataset contains information for MCF-7 and MCF-7/TR cells. These data were analyzed through R software with the affy and limma Bioconductor packages (*Ritchie et al., 2015*). To identify tamoxifen resistance-related DEGs between MCF-7/TR and MCF-7 cells, Student's $t$ test was carried out and statistical significance was set at $P < 0.05$ with a fold change ≥1.5.

### DEG gene ontology and pathway enrichment analysis

To determine the biological functions and signaling pathways related to the DEGs, we submitted the data to Database for Annotation, Visualization and Integrated Discovery (DAVID), which is an online tool for gene annotation, function visualization and large volume data integration, including GO functional analysis and Kyoto Encyclopedia of Genes and Genomes (KEGG) pathway analysis (*Schmid & Blaxter, 2008*; *Huang, Sherman & Lempicki, 2009*). GO classifications include CC, BP and MF groups and $P < 0.001$ was considered significant.

### Protein–protein interaction network construction by STRING

To evaluate protein–protein interaction (PPI) network information for the significant DEGs, we analyzed the 865 DEGs by the online database Search Tool for the Retrieval of Interacting Genes (STRING) to predict interactions among them (*Szklarczyk et al., 2015*). A combined score >0.7 was considered a high confidence score. These DEGs with many associations with other genes have critical relationships in the PPI interaction network. The top 15 upregulated DEGs in the PPI network were visualized through Cytoscape software (version 3.3.0) (*Shannon et al., 2003*). CytoHubba was employed to detect hub proteins according to their associations with other proteins by using Cytoscape software (version 3.3.0) (*Chin et al., 2014*).

## RT-qPCR assay

The primer sequences used were as follows: ATP Citrate Lyase (*ACLY*) forward, 5′-TCGGCCAAGGCAATTTCAGAG-3′ and reverse, 5′-CGAGCATACTTGAACC GATTCT-3′; Heat Shock Protein Family D (Hsp60) Member 1 (*HSPD1*) forward, 5′-ATGCTTCGGTTACCCACAGTC-3′ and reverse, 5′-AGCCCGAGTGAGATGAGG AG-3′; Phosphoribosylformylglycinamidine Synthase (*PFAS*) forward, 5′-CGAG CATACTTGAACCGATTCT-3′ and reverse, 5′-GTAGCACAGTTCAGTCTCGAC-3′; Phosphoribosylglycinamide Formyltransferase (*GART*) forward, 5′-GGAATCCCAACCG CACAATG-3′ and reverse, 5′-AGCAGGGAAGTCTGCACTCA-3′); Thioredoxin (*TXN*) forward, 5′-GTGAAGCAGATCGAGAGCAAG-3′ and reverse, 5′-CGTGGCTGAGAA GTCAACTACTA-3′; Heat Shock Protein Family H (Hsp110) Member 1 (*HSPH1*) forward, 5′-ACAGCCATGTTGTTGACTAAGC-3′ and reverse, 5′-GCATCTAACACA GATCGCCTCT-3′; Heat Shock Protein Family E (Hsp10) Member 1 (*HSPE1*) forward, 5′-ATGGCAGGACAAGCGTTTAGA-3′ and reverse, 5′-TGGAAGCATAATGCCTCC TTTG-3′; TNF Receptor Associated Protein 1 (*TRAP1*) forward, 5′-AGGACGACTGTTC AGCACG-3′ and reverse, 5′-CCGGGCAACAATGTCCAAAAG-3′; Nischarin (*IRAS*) forward, 5′-CAACCT ACGGATGACTCGTGCTT-3′ and reverse, 5′-TTTCTCCCTGAC GGTCCCACTT-3′; 5-Aminoimidazole-4-Carboxamide Ribonucleotide (*ATIC*) forward, 5′-ACCTGACCGCTCTTGGTTTG-3′ and reverse, 5′-TACGAGCTAGGATTCCAG CAT-3′. Glyceraldehyde-3-Phosphate Dehydrogenase (GAPDH) forward, 5′-GGAGCG AGATCCCTCCAAAAT-3′ and reverse, 5′-GGCTGGGTGCATCATTCTCATGG-3′.

## Survival analysis of hub genes

We analyzed the overall survival of breast cancer patients stratified by the expression of 10 hub genes (high and low) using the Kaplan–Meier Plotter (http://www.kmplot.com), which is an online tool accumulating data on the gene expression and survival of breast cancer patients.

## Cell viability assay

The breast cancer cell lines were plated in 96-well plates (2,000–5,000 cells/well) and treated with escalating doses of TAM as single agent. According to the manufacturer's instructions, MTS assays (Promega, Madison, WI, USA) were conducted to detect the proliferation ability of tumor cells. The half maximal inhibitory concentrations (IC50) were calculated by GraphPad Prism version 7. Data were presented as the mean ± SD of three independent experiments.

## siRNA transfection

MCF-7/TR cells were transfected with specific siRNA or negative control siRNA using RNAi MAX and the negative control were designed by ribo. These cells were cultured for 24–48 h in a $CO_2$-containing chamber at 37 °C before being used for cell-based assays.

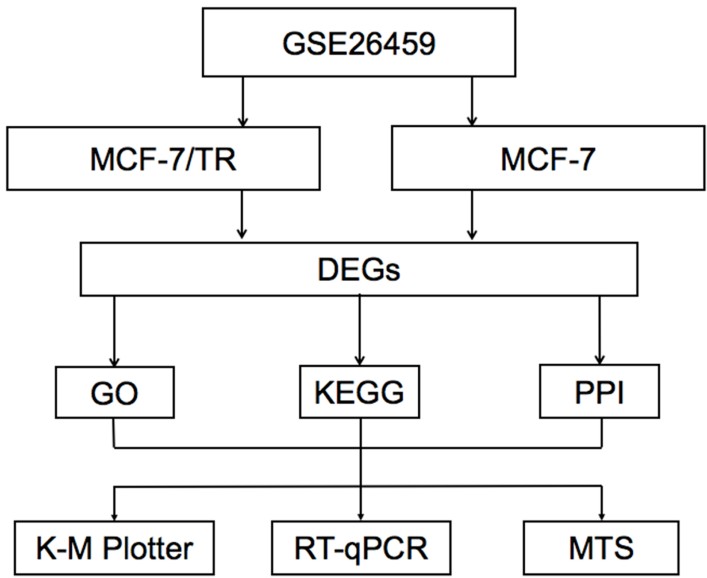

**Figure 1** The flowchart of the bioinformatics analysis DEGs, differentially expressed genes; GO, Gene Ontology; KEGG, Kyoto Encyclopedia of Genes and Genomes; PPI, protein-protein interaction; K–M Plotter, Kaplan–Meier Plotter; RT-qPCR, quantitative real-time polyme.

## Statistical analysis

All data were statistically analyzed using Student's $t$ test with GraphPad Prism 7 software. The internal GAPDH gene was used for normalization of gene expression, at $^*P < 0.05$; $^{**}P < 0.01$; $^{***}P < 0.001$ compared with MCF-7 cells based on Student's $t$ test.

## RESULTS

### Identification of DEGs

The research process of bioinformatics analysis is displayed in Fig. 1. We identified 865 significant DEGs, consisting of 399 upregulated and 466 downregulated DEGs, between MCF-7 cell lines and tamoxifen-resistant MCF-7/TR cell lines based on the public microarray dataset GSE26459. Among them, the most significantly upregulated gene was *WNT2B*, and the most significantly downregulated gene was *ALCAM*. The top ten downregulated and upregulated DEGs are presented in Table 1.

### DEG gene ontology and pathway enrichment analysis

We clustered the DEGs through GO and KEGG pathway analyses in DAVID to explore potential biological roles and functional enrichment. The enriched GO terms, including BP, CC and MF, are presented in Figs. 2A–2C, respectively. The most significantly enriched BP GO terms were "telomere organization" (GO:0032200), "nucleosome assembly" (GO:0006334) and "chromatin silencing at rDNA" (GO:0000183) (Fig. 2A). In the CC category (Fig. 2B), cell membrane components were the major enriched categories, which included extracellular exosomes (173 genes), protein complexes (41 genes) and extracellular matrices (34 genes). Other enriched GO terms consisted of

**Table 1 The most significant up-regulated and down-regulated DEGs (top ten, Tamoxifen-resistant MCF-7/TR vs Tamoxifen-sensitive MCF-7).**

| Gene | logFC | P Value | Gene full name | Gene ID number |
|---|---|---|---|---|
| **Up-regulated** | | | | |
| WNT2B | 5.315 | $7.3 \times 10^{-12}$ | Wnt Family Member 2B | 7482 |
| EVL | 5.292 | $4.1 \times 10^{-11}$ | Enah/Vasp-Like | 51466 |
| TBC1D9 | 2.331 | $1.1 \times 10^{-10}$ | TBC1 Domain Family Member 9 | 23158 |
| LMCD1 | 2.862 | $1.1 \times 10^{-10}$ | LIM And Cysteine Rich Domains 1 | 29995 |
| CSTA | 3.675 | $1.5 \times 10^{-10}$ | Cystatin A | 1475 |
| SLC6A14 | 3.391 | $5.9 \times 10^{-10}$ | Solute Carrier Family 6 Member 14 | 11254 |
| PODXL | 2.196 | $7.1 \times 10^{-10}$ | Podocalyxin Like | 5420 |
| COX6C | 2.163 | $1.2 \times 10^{-09}$ | Cytochrome C Oxidase Subunit 6C | 1345 |
| TNS3 | 2.431 | $1.3 \times 10^{-09}$ | Tensin 3 | 64759 |
| ACSL1 | 2.406 | $1.3 \times 10^{-09}$ | Acyl-CoA Synthetase Long Chain Family Member 1 | 2180 |
| **Down-regulated** | | | | |
| ALCAM | −5.393 | $3.7 \times 10^{-12}$ | Activated Leukocyte Cell Adhesion Molecule | 214 |
| LXN | −4.032 | $7.3 \times 10^{-12}$ | Latexin | 56925 |
| TARP | −5.160 | $2.1 \times 10^{-11}$ | TCR Gamma Alternate Reading Frame Protein | 445347 |
| TRGC2 | −5.409 | $3.7 \times 10^{-11}$ | T Cell Receptor Gamma Constant 2 | 6967 |
| PPP3CA | −2.883 | $1.8 \times 10^{-10}$ | Protein Phosphatase 3 Catalytic Subunit Alpha | 5530 |
| CLEC7A | −2.609 | $2.8 \times 10^{-10}$ | C-Type Lectin Domain Containing 7A | 64581 |
| SOWAHA | −2.190 | $5.8 \times 10^{-10}$ | Sosondowah Ankyrin Repeat Domain Family Member A | 134548 |
| CD36 | −3.636 | $6.3 \times 10^{-10}$ | CD36 Molecule | 948 |
| CLEC3A | −2.534 | $6.8 \times 10^{-10}$ | C-Type Lectin Domain Family 3 Member A | 10143 |
| SDC2 | −2.583 | $8.2 \times 10^{-10}$ | Syndecan 2 | 6383 |

membrane (124 genes) and cell-cell adherens junctions (33 genes, Fig. 2B). In the MF category (Fig. 2C), the most important GO categories were binding-related items for protein heterodimerization activity (60 genes) and histone binding (29 genes, Fig. 2C).

Moreover, KEGG pathway analysis showed enrichment of five key pathways, including antigen processing and presentation signaling (8 genes), allograft rejection (6 genes) and cell adhesion molecules (CAMs) (11 genes, Table 2).

Hub genes in STRING analysis of tamoxifen resistance were used to identify the PPI of the DEGs. PPI information for the 865 DEGs was evaluated in the STRING website, and we then used Cytoscape software to establish the PPI network for the top 15 DEGs, which contained 108 nodes and 226 edges.

The hub genes were evaluated by CytoHubba. In the PPI networks, 10 node proteins—ACLY, HSPD1, PFAS, GART, TXN, HSPH1, HSPE1, IRAS, TRAP1 and ATIC—were associated strongly with other node proteins (more than 5), suggesting a high degree of connectivity (Fig. 3). Consistently, RT-qPCR analysis indicated that the expression level of these 10 genes was increased in MCF-7/TR (tamoxifen-resistant cell line) cells comparing with MCF-7 (tamoxifen-sensitive cell line) cells (Fig. 4). Collectively, it's indicated that these hub genes and proteins may play a critical role in tamoxifen resistance.

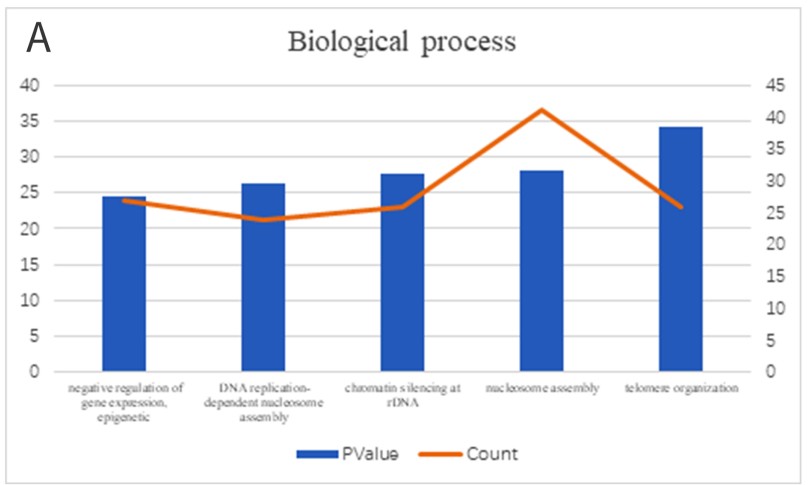

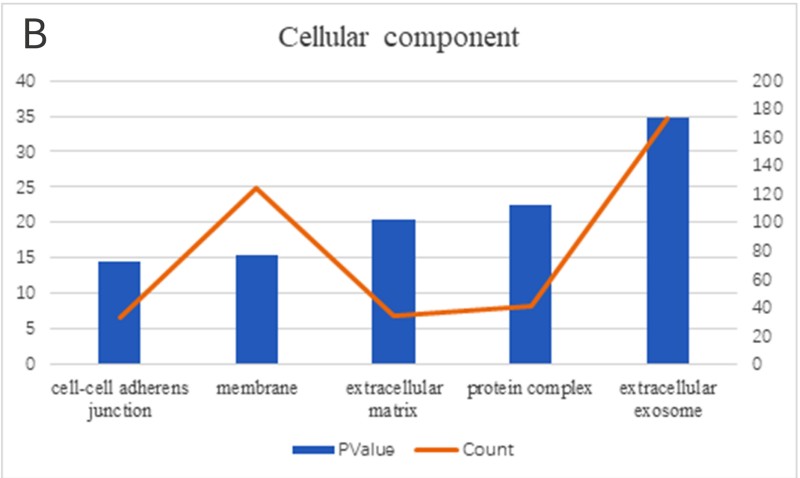

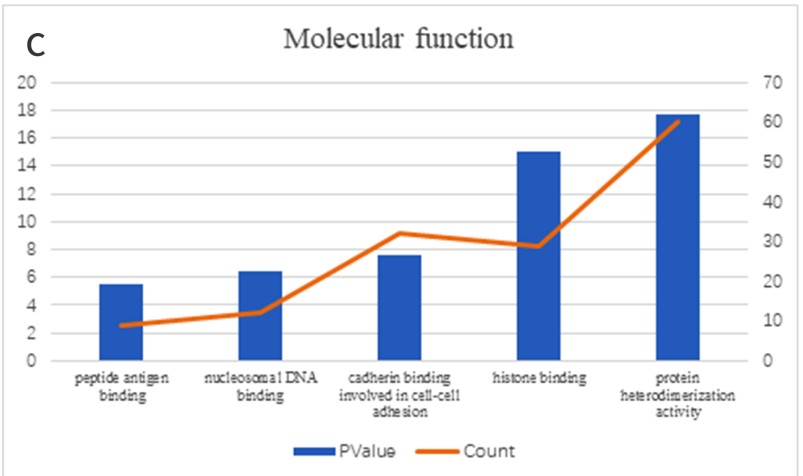

**Figure 2 GO enrichment of DEGs.** (A) In biological process ontology; (B) in cellular component ontology; (C) in molecular function ontology.

**Table 2 The enriched KEGG pathway of DEGs.**

| Term | Description | Count | P Value |
|---|---|---|---|
| hsa04612 | Antigen processing and presentation | 8 | 9.71E−04 |
| hsa05330 | Allograft rejection | 6 | 9.34E−04 |
| hsa04514 | Cell adhesion molecules (CAMs) | 11 | 7.19E−04 |
| hsa05332 | Graft-vs-host disease | 6 | 5.44E−04 |
| hsa04940 | Type I diabetes mellitus | 7 | 2.09E−04 |

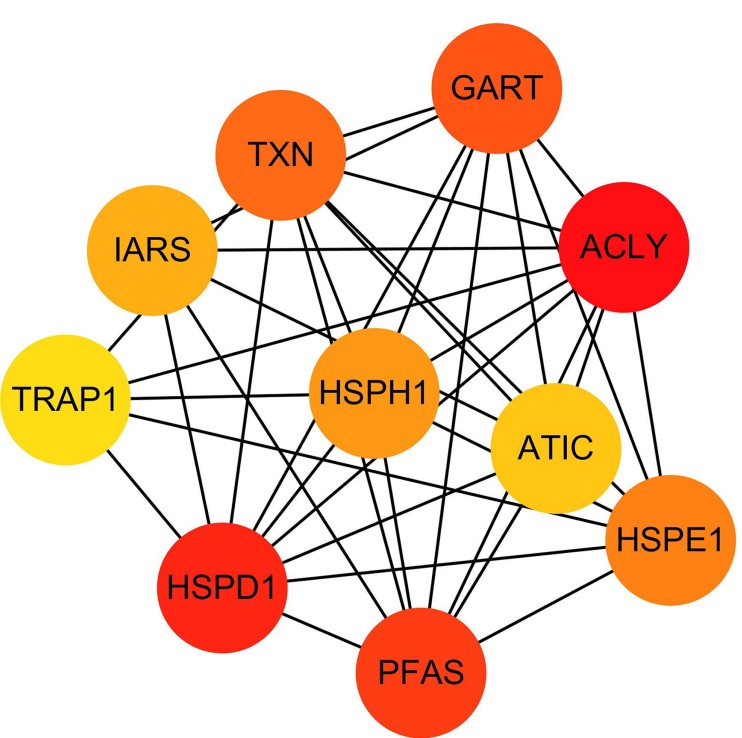

**Figure 3 The PPI network diagram of the node proteins showed many associations with other node proteins, indicating that they have high degrees of connectivity.** The darker the color was, the more proteins associated with the node protein.

## Tamoxifen resistance-associated hub genes predict poor prognosis

For survival analysis, we evaluated associations between expression of the 10 hub genes and patient survival with the Kaplan–Meier Plotter database. Four hub genes (*TXN, HSPD1, HSPH1* and *ATIC*) were found to be related to poor overall survival in breast cancer patients who received tamoxifen (Fig. 5).

## Knockdown of target genes by siRNAs

MTS assay verified that MCF-7/TR cell line (IC50 17.21μM) was less sensitive to tamoxifen than MCF-7 cell line (IC50 7.46μM) ($P < 0.05$) (Fig. 6A). RT–qPCR experiments confirmed that small interfering RNA (siRNA) resulted in over 60% reduction in expression of 10 hub genes (Fig. 6B). MTS assay showed that knockdown of HSPH1

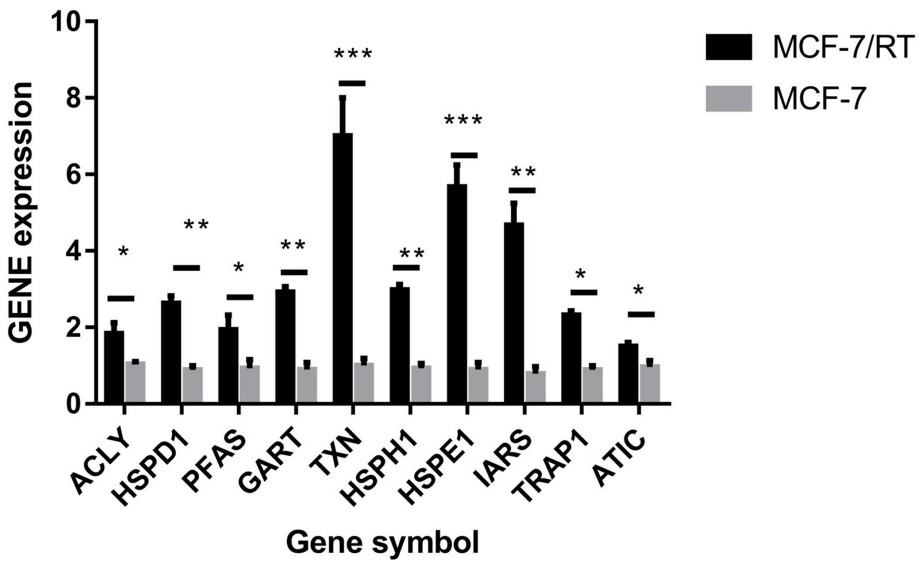

**Figure 4 RT-qPCR validation of the DEGs between MCF-7 and MCF-7/TR.** Relative expression of ACLY, HSPD1, PFAS, GART, TXN, HSPH1, HSPE1, IRAS, TRAP1, and ATIC. *$P < 0.05$; **$P < 0.01$; ***$P < 0.001$.

may lead to reduced growth of MCF-7/TR cell ($P = 0.07$), whereas knockdown of other genes did not significantly affect cell growth (Fig. 6C). It's indicated that upregulation of HSPH1 is likely to play an important role in tamoxifen resistance.

## DISCUSSION

Tamoxifen resistance is an unsolved problem in breast cancer treatment, hence it is particularly important to identify critical genes and signaling pathways related to tamoxifen resistance to determine its complex mechanism. In the present study, we identified 865 DEGs, including 399 upregulated and 466 downregulated DEGs, between the MCF-7 and MCF-7/TR cell lines. These DEGs possibly mediated tamoxifen resistance. Previous findings suggested that several DEGs might be responsible for resistance in diverse cancers. For example, *WNT2B*, among the top ten upregulated DEGs, is important in chemotherapy resistance and tumorigenesis in head and neck squamous cell carcinoma and may be a treatment target for oral cancers (*Li, Yang & Qian, 2015*). The present study identified *SLC6A14* as a new treatment target for ER-positive breast cancer. It's indicated that *SLC6A14* may play a role in tamoxifen resistance (*Karunakaran et al., 2011*). *COX6C*, listed in the top ten upregulated DEGs, affects mitochondrial genes and function and induces *ABCG2*-overexpressing cells to pump mitoxantrone from the cell matrix (*Chang et al., 2017*). *ALCAM*, among the top ten downregulated DEGs, is a new serum marker correlating with cell growth, adhesion and chemotherapy resistance in pancreatic cancer (*Hong et al., 2010*). In addition, the miR-20a-5p/SDC2 axis is a possible biomarker for diagnosis and treatment targets (*Zhao et al., 2017*). In conclusion, these DEGs are likely to play vital roles in tamoxifen resistance via various mechanisms.

We explored the biological functions of the DEGs with GO analysis. In the BP category, the DEGs were mostly enriched in telomere organization; other DEGs were

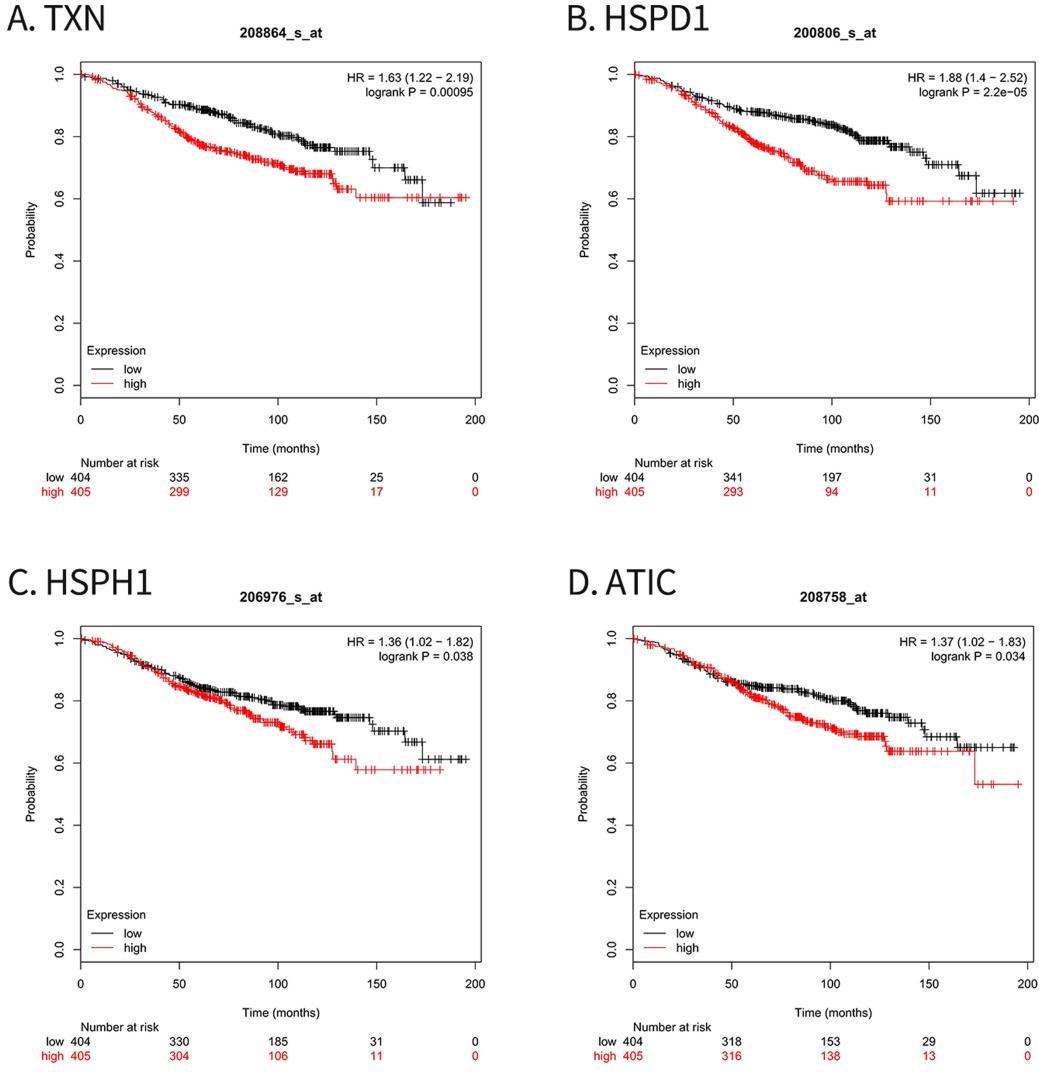

**Figure 5  Kaplan–Meier Plotter analysis of four genes involved in breast cancer.** (A) TXN; (B) HSPD1; (C) HSPH1; (D) ATIC. HR, Hazard ratio; CI, confidence interval.

enriched in nonmembrane-related items, including nucleosome assembly and chromatin silencing at rDNA. Regarding the CC category, stimulus-related items, including extracellular exosomes, were most significant. Other enriched categories included protein complexes and extracellular matrix. These data showed that the molecular mechanisms of tamoxifen resistance might be related to both membrane and nonmembrane structures. Binding-related items for protein heterodimerization activity were most significant in the MF category, and other enriched MF terms included histone binding and nucleosomal DNA binding.

Analysis of pathways may uncover more exact biological functions of DEGs than those obtained through Gene Ontology analysis. We identified three enriched pathways: antigen processing and presentation signaling pathways, pathways in allograft rejection and CAMs. Cancer pathways may participate in cancer evolution, including drug resistance.

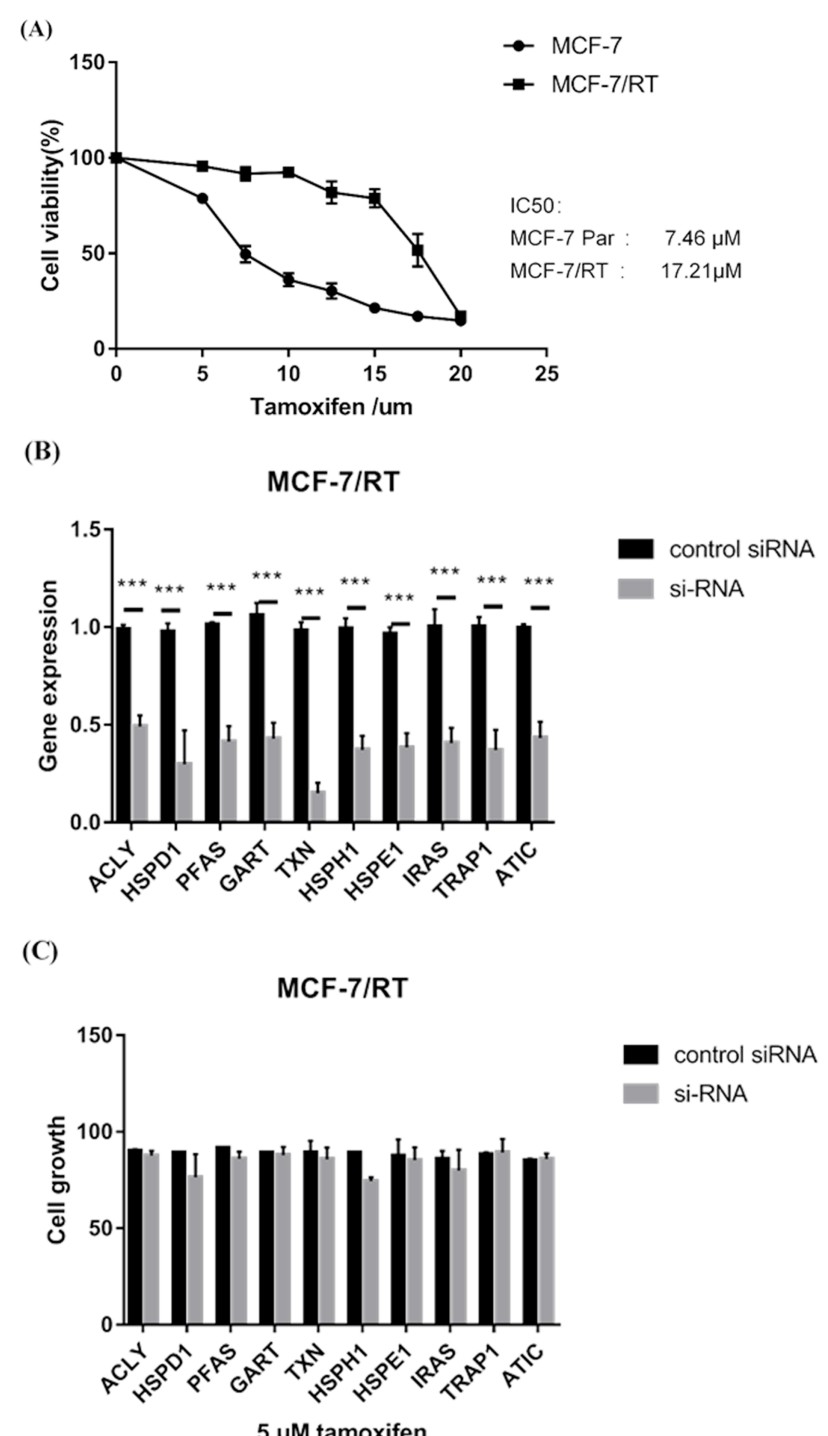

**Figure 6** (A) MTS assay verified that MCF-7/TR cell line (IC50 17.21μM) was less sensitive to tamoxifen than MCF-7 cell line (IC50 7.46 μM) (P < 0.05). (B) RT–qPCR experiments confirmed that small interfering RNA (siRNA) resulted in over 60% reduction in expression of 10 hub genes. (C) MTS assay showed that knockdown of HSPH1 may lead to reduced growth of MCF-7/TR cell with a trend close to significance (P = 0.07), whereas knockdown of other genes did not significantly affect cell growth.

Our PPI network included *ACLY, HSPD1, PFAS, GART, TXN, HSPH1, HSPE1, IRAS, TRAP1,* and *ATIC*, which were confirmed by RT-qPCR, and several of them are associated with drug resistance. Four hub genes (*TXN, HSPD1, HSPH1* and *ATIC*) were related to OS in patients who accepted tamoxifen treatment according to Kaplan–Meier Plotter. It's indicated that the overexpression of TXN proteins was one of the atypical multidrug resistance mechanisms associated with overexpression of P-glycoprotein (*Mieszala et al., 2018*). TXN modulate the expression of estrogen responsive genes in breast cancer cells, and reducing oxidized TXN restore tamoxifen sensitivity to previously resistant breast cancer cells (*Rosalind & Deodutta, 2013*). A previous study demonstrated that the expression level of the HSPD1 mRNA was closely related to the emergence of cisplatin resistance in head and neck cancer cell lines (*Nakata et al., 1994*). In addition, high transcript level of HSPD1 was associated with the resistance to hormonal therapy, and a short 4-gene signature including HSPD1 might effectively predict tamoxifen-resistance (*Sotgia, Fiorillo & Lisanti, 2017*). The function of HSPH1 in DNA repair may partially explain the higher resistance to genotoxic drugs of colorectal cancers expressing higher levels of HSPH1 (*Causse et al., 2018*). ATIC was associated with resistance to methotrexate (*You et al., 2013*). We further knocked down 10 hub genes by siRNAs and found silence of HSPH1 led to reduced growth of MCF-7/TR cell with a tendency toward statistical significance ($P = 0.07$). Upregulation of HSPH1 may be important in the mechanism of tamoxifen resistance.

Several limitations of the present study merit discussion, including the fact that only the GEO26459 dataset of microarray data was used and that the sample size was relatively small. The aforementioned results, including the DEGs and their functions, should be confirmed by in vivo or in vitro experiments, which will be carried out in future studies. Nonetheless, the preliminary findings of our study provide comprehensive molecular insight and potential directions for further elucidating the underlying mechanisms of tamoxifen resistance.

# CONCLUSIONS

Our study revealed a number of critical hub genes which might play an important role in the mechanism of tamoxifen resistance. These hub genes, including *TXN, HSPD1, HSPH1,* and *ATIC*, serve as potential therapeutic targets in breast cancer resistant to tamoxifen and suggest directions for uncovering the mechanisms of tamoxifen resistance. The study results have provided a theoretical basis for future translational research, which may eventually solve the clinical problem of tamoxifen resistance.

## Funding
The authors received no funding for this work.

## Competing Interests
The authors declare that they have no competing interests.

## Author Contributions

- Kai Zhang conceived and designed the experiments, performed the experiments, analyzed the data, prepared figures and/or tables, authored or reviewed drafts of the paper, and approved the final draft.
- Kuikui Jiang conceived and designed the experiments, performed the experiments, analyzed the data, prepared figures and/or tables, authored or reviewed drafts of the paper, and approved the final draft.
- Ruoxi Hong conceived and designed the experiments, analyzed the data, prepared figures and/or tables, and approved the final draft.
- Fei Xu conceived and designed the experiments, analyzed the data, prepared figures and/or tables, and approved the final draft.
- Wen Xia conceived and designed the experiments, analyzed the data, prepared figures and/or tables, and approved the final draft.
- Ge Qin performed the experiments, prepared figures and/or tables, and approved the final draft.
- Kaping Lee performed the experiments, prepared figures and/or tables, and approved the final draft.
- Qiufan Zheng performed the experiments, prepared figures and/or tables, and approved the final draft.
- Qianyi Lu performed the experiments, prepared figures and/or tables, and approved the final draft.
- Qinglian Zhai performed the experiments, prepared figures and/or tables, and approved the final draft.
- Shusen Wang conceived and designed the experiments, authored or reviewed drafts of the paper, and approved the final draft.

## Data Availability

The raw data are available in the Supplemental Files.

## Supplemental Information

Supplemental information for this article can be found online at http://dx.doi.org/10.7717/peerj.10468#supplemental-information.

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
