# Peer review of "Identification and characterization of critical genes associated with tamoxifen resistance in breast cancer"

_PeerJ, doi:10.7717/peerj.10468_

## Round 0.1 · original submission · Major Revisions

For further consideration, the authors must carefully and satisfactorily implement all referee comments.

Reviewer 1 ·

Basic reporting

Zhang and coworkers investigated the potential mechanism of tamoxifen resistance in this manuscript entitled “Identification and characterization of crucial genes associated with tamoxifen resistance in breast cancer”. They utilized the already established bioinformatics approaches to re-analysis of microarray data from a public repository with accession no. GSE26459. The bioinformatics analysis demonstrated the candidate hub genes, gene ontologies, and pathways to discuss the tamoxifen resistance. The typical analysis leads to a set of prognostic hub genes. I commend the immunohistochemical assay showed 2 hub proteins as a significant drug target in tamoxifen resistance in breast cancer.

Experimental design

no comment

Validity of the findings

no comment

Additional comments

Some suggestions listed below for consideration by the authors to improve the manuscript:

1) Please clarify in the introduction how your work is different than previous works. I would suggest to include a few more sentences on line 97 to clarify the contributions of your work compared to previous work.
2) In the methods section, gene ontology and pathways analysis (line 138), could you please mention which statistical test was utilized to screen out the statistically significant terms?
3) In the method section, In PPI analysis (line 145), why you considered only the top 15 as hubs? What was the analysis methods of the PPI and parameters?
4) In Table 1, could it be possible to write the log FC in 2/3 decimal and scientific notation can be used for P-value instead of E notation?
5) I wonder why the authors only consider KEGG pathways, while there are other 6-7 data sources such as Reactome, NCI, etc. I would suggest, since the paper mostly discuss the pathway-based mechanism of tamoxifen resistance, to incorporate other pathways annotations in this manuscript.
5) I think Figure 1 is a very simplistic diagram with abbreviations, and adding much description in the legend would help the readers.
6) The resolution of fig 2 is not clear.
7) The survival analysis part lacks methodological descriptions.
8) The roles of hub genes can be explained in more detail in the discussion.

·

Basic reporting

(1) Some descriptions of this paper are unambiguous. Please see Major Q1.
(2) Literature well referenced and relevant.
(3) Figures are low quality. Please see Minor Q1 to Q6.
(4) Raw data are supplied.

Experimental design

(1) The experimental method has some problems and needs to be redesigned. Please see Major Q4 and Q7.

Validity of the findings

(1) More research data are needed to support the conclusion of this study. Please see Major Q1 to Q7.

Additional comments

Breast cancer is one of the predominant types of tumors in women worldwide. Despite great progress in the early diagnosis and treatment recently, drug resistance and distal metastasis remain major causes of cancer-related mortality. In this manuscript, Kai Zhang and colleagues investigated the potential mechanisms of tamoxifen resistance through bioinformatics analysis. The study revealed a number of critical hub genes that might serve as therapeutic targets in breast cancer resistant to tamoxifen and provided potential directions for uncovering the mechanisms of tamoxifen resistance. Although, the current study is interesting but there are some major concerns that need to be addressed.

Major concerns:
1. “Title” “crucial genes” The present study only screened the differentially expressed genes associated with tamoxifen resistance and did not prove that the screened genes were involved in tamoxifen resistance. This manuscript contains a lot of loose language, which exaggerates the significance of the relevant results and must be carefully revised by the author.
2. “Cell culture” Line 125-130. Because the cells in this study were not purchased directly, the authors should provide STR identification results for the two cells.
3. “Immunohistochemistry (IHC)” Line 183-184. Please give a clear definition of endocrine therapy resistance and references.
4. “Statistical analysis” Line 194-196. The IHC clinical analysis lacked statistical methods.
5. “DEG Gene Ontology and pathway enrichment analysis” Line 228-231. In addition to expression verification, the authors should use siRNA to silence the target gene and observe whether breast cancer cells can sensitive to tamoxifen. This is a simple and low cost experiment, but it is important to support the conclusions of this study and should be presented by the authors.
6. “DEG Gene Ontology and pathway enrichment analysis” Line 228-231. Some genes should measure protein levels. In particular, two genes were tested for IHC.
7. “HSPD1 and HSPH1 levels correlate with tamoxifen resistance” Line 238-250. In tamoxifen-resistant patients, the comparison of expression differences between ANT and tumor tissues is not an important data supporting the conclusions of this study. The authors should compare the expression between tumor tissues of patient’s sensitive to endocrine therapy and those insensitive to endocrine therapy, or compare the expression between tumor samples sensitive to endocrine therapy and tumor samples resistant to endocrine therapy of the same patient. Since the author has collected 50 samples of endocrine therapy resistance from 2001 to 2016, I believe that the author is sure to get more samples of patient’s sensitive to endocrine therapy.

Minor concerns:
1. Table 1. Just keep 3 decimal places. List the probe number, gene full name, and gene ID number.
2. Figure 1. Flowcharts should not only show the process of bioinformatics analysis, but should present the entire research process.
3. Figure 2. Please make sure the text in the picture is large and clear for the author to read. The title of the axes should be clear.
4. Figure 3. The nodes in the diagram need to distinguish significantly between the DEGs and other genes. And add color legends.
5. Figure 5. Please make sure the text in the picture is large and clear for the author to read. To avoid direct screenshots, use vector images.
6. Figure 6. The control tissue in Figure 1B does not look like normal breast tissue. The author should check it with a pathologist.

·

Basic reporting

Figure 2: Why do A and C lack the detailed of the number of genes in the graphs, whereas they are shown in B?
Figure 3: the interaction graph should be improved, spread out further to be able to appreciate all the interactions.
Table 4: the layout should be improved, it is confusing and not intuitive.
Figure 6: more IHC should be shown (at least 3 patients for each protein). Also, in the case of Patient D the normal tissue does not look like normal mammary gland. Is it stroma? Could a more representative section be chosen?

Experimental design

The experimental design is adequate.

Validity of the findings

The paper is valid. The authors should explain why they chose a threshold of 1.5 for the DEG. The conclusions are adequate.

Additional comments

The paper entitled: Identification and characterization of crucial genes 3 associated with tamoxifen resistance in breast cancer, by Kai Zhang, Kuikui Jiang, Ruoxi Hong, Fei Xu, Wen Xia, Ge Qin, Kaping Lee, Qiufan Zheng, Qianyi Lu, Qinglian Zhai, Shusen Wang compares gene expression profiles of MCF7 and MCF7/TR cells using a GEO dataset. Gene Ontology (GO) and Kyoto Encyclopedia of Genes and Genomes (KEGG) pathway enrichment analyses using Database for Annotation, Visualization and Integrated Discovery (DAVID). A protein-protein interaction (PPI) network is generated, hub genes were analyzed. Finally, the authors conducted the immunohistochemical analysis of samples from breast cancer patients.
The paper is clearly written. The following issues should be considered:
1) The authors should explain why they chose a threshold of 1.5 for the DEG.
2) Figure 2: Why do A and C lack the detail of the number of genes in the graphs, whereas they are shown in B?
3) Figure 3: the interaction graph should be improved, spread out further to be able to appreciate all the interactions.
4) Table 4: the layout should be improved, it is confusing and not intuitive.
5) Figure 6: more IHCs should be shown (at least 3 patients for each protein). Also, in the case of Patient D the normal tissue does not look like normal mammary gland. Is it stroma? Could a more representative section/area be chosen?

---

## Round 0.2 · accepted · Accept

The authors successfully implemented all reviewers' recommendations and comments. I congratulate the authors for their work.

Reviewer 1 ·

Basic reporting

no comment

Experimental design

no comment

Validity of the findings

no comment

Additional comments

I think the authors have improved the manuscript significantly and answered my criticisms satisfactorily.

·

Basic reporting

No comment.

Experimental design

No comment.

Validity of the findings

No comment.

Additional comments

Breast cancer is one of the predominant types of tumors in women worldwide. Despite great progress in the early diagnosis and treatment recently, drug resistance and distal metastasis remain major causes of cancer-related mortality. In this manuscript, Kai Zhang and colleagues investigated the potential mechanisms of tamoxifen resistance through bioinformatics analysis. The study revealed a number of critical hub genes that might serve as therapeutic targets in breast cancer resistant to tamoxifen and provided potential directions for uncovering the mechanisms of tamoxifen resistance. The paper is improved and most concerned raised by the reviewer have been addressed. I think it is might suitable for publication at this version of revised manuscript.